# Hyaluronan Oligosaccharides-Coated Paclitaxel-Casein Nanoparticles with Enhanced Stability and Antitumor Activity

**DOI:** 10.3390/nu14193888

**Published:** 2022-09-20

**Authors:** Man Wang, Yahui Zhang, Zuqi Fei, Dongchao Xie, Haihua Zhang, Qizhen Du, Peng Jin

**Affiliations:** Department of Food Science and Technology, College of Food and Health, Zhejiang A & F University, Hangzhou 311300, China

**Keywords:** hyaluronic acid oligosaccharide, bovine casein, encapsulation, drug delivery, paclitaxel

## Abstract

This study aims to develop specific-molecular-weight hyaluronic acid oligosaccharides-coated paclitaxel-loaded casein nanoparticles (HA-PT-Cas NPs) via chemical conjugation to increase the stability and antitumor effects. Optimized HA-PT-Cas NPs (HA/casein of 3:1) were obtained with a mean size of 235.3 nm and entrapment efficiency of 93.1%. HA-PT-Cas exhibited satisfactory stability at 4 °C for 12 days and 37 °C for 3 h; paclitaxel was retained at rates of 81.4% and 64.7%, respectively, significantly higher than those of PT-Cas (only 27.8% at 4 °C after 16 h and 20.3% at 37 °C after 3 h). HA-PT-Cas exhibited high efficiency (61.3%) in inhibiting A375 tumor owing to the enhanced stability of HA oligosaccharides barrier, which was comparable with that of 10 μg/mL cis-platinum (64.9%). Mice experiments showed the 74.6% tumor inhibition of HA-PT-Cas by intravenously administration, significantly higher than that of PT-casein (39.8%). Therefore, this work provides an effective carrier for drug delivery via HA oligomers-coated modification.

## 1. Introduction

Casein, as the main component of bovine milk protein, is widely used in food processing owing to its excellent bioresorbability, biocompatibility and biodegradability. Natural caseins have self-assembly ability, as well as good emulsifying and amphiphilic properties, thereby forming the stable micellar structure in solution with diameters in the approximate range of 50–500 nm. Considering their excellent properties, caseins have potential for use as promising nanocarriers for hydrophobic drug delivery systems. For example, casein was used to encapsulate insoluble drugs such as celecoxib [1], hydrochlorothiazide [2], and triptolide [3] nanocomposites, significantly increasing drugs maintained-release rates and bioavailability by 1.5- to 49-fold. However, the internal structure of micelles is porous, and forms a large number of cavities with irregular channels (d > 5 nm) and inner cavities (d = 20–30 nm) [4]. This feature serves as a reservoir in to which drug molecules depending on their physicochemical properties could be incorporated by means of physical, chemical, or electrostatic interactions. Although the former micelles bestow caseins powerful loading capacity for drug encapsulation, the loaded hydrophobic drugs usually show low stability and fast leakage due to the lack of block outside the channels and cavities. Therefore, this feature is critical for improving polymer micelles as drug delivery systems to enhance the stability of drug-loaded casein nanoparticles (NPs) by the structural modification of micelle surface [5].

Recently, some new polymer materials, such as polyethylene glycol [6], chitosan [7,8], mannans [9], and hyaluronic acid [10,11], have been widely used as the block copolymer to coat and modify the surface of NPs. These nanocomplexes with block copolymer modifications have higher stability and bioavailability of hydrophobic drug molecules. Hyaluronic acid, a linear and unbranched high-molecular weight (HMW) glycosaminoglycan, is widely distributed in various host tissues and participates in crucial physiological processes [12]. In particular, low-molecular-weight HA oligosaccharides have unique biological activities in cell motility, inflammation, wound healing, and cancer metastasis, and was specifically recognized and bound by CD44 on the surface of tumors [13]. HA is an endogenous substance in the human body and has good biocompatibility and biodegradability, especially with the ubiquitous hyaluronidase in tumor tissues; the hydrolysis of HA in due course makes nanocomplexes efficiently release drug molecules at targeted sites [14]. Thus, HA oligosaccharides have a greater potential for the modification of nano-delivery system and encapsulation of drug molecules.

Owing to the presence of several chemical groups (hydroxyl and carboxylic acid) on HA backbone, HA have been successfully utilized in drug chemical conjugation creating a macromolecular prodrug [15]; the conjugated drug becomes active upon release from the HA. Unfortunately, the uptake of HA-conjugated drug in vivo could be blocked by both excess HA and anti-CD44 antibodies, and the chemically conjugated drug molecule is released by intracellular enzymatic hydrolysis resulting in uncontrollable release. The HMW-HA liposomes were originally proposed to be used as a bioadhesive formulation attached to a nanocarrier via electrostatic attraction and as a protective structural targeting coating for drug delivery [14]. However, such simple physical electrostatic interactions make the stability of nanoparticles in storage or in application environment difficult to maintain.

Paclitaxel (PT) has shown tremendous potential as powerful antimitotic agent in cancer therapy, however, expanding the applications of PT is compromised by poor aqueous solubility and by essential solubilization Cremophor of its commercial formulation. Therefore, in this study, the excellent amphiphilic and self-assembly properties of casein micelles were employed to encapsulate high-hydrophobicity paclitaxel drug molecules for the formulation of paclitaxel casein NPs. Subsequently, the exposed surface reactive amino groups (lysine ε amino and terminal amino) of casein NPs as effective binding sites for chemical conjugation with the hydroxyl groups of specific molecular weight HA oligosaccharides, linear HA was firmly adhered to the surface of casein-paclitaxel nanoparticles, thereby forming a stable copolymeric nanocomplexes. Thus, the stability and biological activity of casein-paclitaxel nanoparticles significantly improved. 

## 2. Materials and Methods

### 2.1. Materials and Reagents

Bovine casein (>98%), 1-(3-dimethylaminopropyl)-3-ethylcarbodiimide hydrochloride (EDC) and N-hydroxysuccinimide (NHS) were obtained from Sigma–Aldrich, Chemical Co. (St. Louis, MO, USA). Paclitaxel (>99%) and cis-diammineplatinum dichloride were purchased from Shanghai Macklin Biochemical Technology Co., Ltd. (Shanghai, China). Human malignant melanoma cells (A375) were purchased from Shanghai Institutes for Biological Sciences (Chinese Academy of Sciences). All materials for cell culture were obtained from Sangon Biotech (Shanghai) Co., Ltd., Shanghai, China. The preparation of specific-molecular-weight HA oligosaccharides (10,000 Da) was performed by using the methods mentioned in our previous study [12,16]. High-molecular-weight HA was hydrolyzed by leech hyaluronidase, and the molecular weight of hydrolyzed HA was determined by high performance size exclusion chromatography. All other reagents were of analytical grade.

### 2.2. Preparation of NPs

Bovine casein was fully dissolved in PBS buffer to achieve a final concentration of 50 mg/mL. Paclitaxel (10 mg) was added into 250 μL anhydrous ethanol, and then dissolved in a vortex mixer for 3 min to form a final concentration of 40 mg/mL. For PT-Cas NP preparation, the casein solution was added into a 2.5 mL PBS buffer with different dilutions to obtain various final concentrations of 10, 20, 30, 40, and 50 mg/mL. The preparation was then placed in a magnetic stirrer with the agitating rate of 500 rpm. Subsequently, paclitaxel solutions were added into the casein solution by inches with various volumes in the range of 40–250 μL, resulting in the final paclitaxel concentrations of 0.64, 1.6, 2.4, 2.8, 3.2, and 4.0 mg/mL. Thereafter, the mixture stirred continuously for 30 min until a paclitaxel-casein nanoparticle (PT-Cas NPs) dispersion was finally generated.

The HA oligosaccharides were fully dissolved in PBS buffer in 50 mL shake flask to form 100 mg/mL of HA solution, which was placed in a magnetic stirrer at an agitating rate of 500 rpm. Subsequently 40 mg EDC and 24 mg NHS were added into the HA solution. The shake flask was sealed and agitated 4 h for hydroxyl activation of HA. After activation, the mixtures were centrifuged by 10,000 rpm at 4 °C for 5 min, and the supernatant was adjusted pH to 8.4. For HA-PT-Cas NPs preparation, the activated HA was added into paclitaxel-casein NP with various ratios of HA: Casein. After the addition, the mixture was sealed and stirred for 2 h at 500 rpm, and then, the insoluble substance was removed through centrifugation at 10,000 rpm for 10 min. Finally, the HA-PT-Cas NP dispersion was obtained. The samples passed through a 0.45 μm filter and stored in at 4 °C for further study.

### 2.3. Characterization of Paclitaxel-Loaded Casein NPs

The physicochemical parameters NP size, polydispersity indexes and zeta potential of nanoparticles were measured as previously descripted [17,18]. Briefly, the NPs suspension was diluted by 10-fold with deionized water. Zetasizer ZS 90 instrument (Malvern Instruments, Malvern, UK) was employed to analyze these parameters in dynamic light scattering with a nominal 5 mW He−Ne laser at 633 nm and 173° scattering angle at 25 °C. The morphological structure of paclitaxel NPs was scanned with Transmission Electron Microscopy (TEM) (Hitachi, H-9500E). For Fourier Transform Infrared (FT-IR) spectra analysis, the dried samples (1 mg) of casein, HA oligomers and HA-PT-Cas were pressed into pellets with KBr powder and then used to analyse over a frequency range of 4000–400 cm^−1^ (650 FT-IR instrument, Tianjin Gangdong Sci. & Tech. Development Co., Ltd., Tianjin, China).

### 2.4. Stability Characterization of NPs

To investigate the effects of HA coating modifications on the stability of the paclitaxel-loaded casein NPs, the samples PT-Cas and HA-PT-Cas nanoparticles with different amounts of HA (HA/Cas 3:1 and 4:1) were incubated at 4 °C and 37 °C, respectively. The entrapment efficiency (EE) of these samples was then measured at predetermined time intervals to evaluate the stability of NPs. A quantitative analysis of paclitaxel was performed by HPLC system equipped with a SPD-M10Avp UV detector (Shimadzu, Kyoto, Japan) and a C18 column (150 mm × 2.0 mm; Shimadzu, Kyoto, Japan). Briefly, 50 μL of the nanosample solution was extracted with 950 μL of methanol, and then diluted to an appropriate amount and the injection volume was 10 μL. The mobile phase consists of acetonitrile: water (*v*/*v* = 55:45) and elute with a constant flow rate of 0.5 mL/min. Meanwhile, the samples PT-Cas and HA-PT-Cas (HA/Cas = 3:1) were lyophilized and stored at room temperature for 30 days. Samples were periodically withdrawn and then re-dissolved into an equal volume of PBS buffer. The nano-particle size and paclitaxel EE were further determined. The experiments were repeated three times.

### 2.5. Cell Viability Assay

Tumor cells were cultured at in Dulbecco’s Modified Eagle’s Medium (DMEM) (GIBCO, Thermo Fisher, Beijing, China) as monolayer cultures under a humidified atmosphere of 95% air and 5% CO_2_ at 37 °C. The culture was supplemented with 10% fetal bovine serum (Thermo Fisher, China), 100 U/mL penicillin and 100 µg/mL streptomycin. 

The effects of HA oligosaccharides modified PT-Cas NPs on the viability of A375 tumor cells were determined by MTT assay as described previously [5]. After cultured for 24 h in in 96-well plates (5000 cells per well), the indicated doses (20 μg/mL of PT and 10 μg/mL of cis-platinum) of samples were used to treat the precultured tumor cells for 48 h. Subsequently, cells were collected, washed twice with PBS buffer, and then incubated in 5% FBS-containing medium at 37 °C for 4 h with 0.5 mg/mL as the final concentration of MTT reagent. After removed the medium of each well, 150 μL of dimethyl sulfoxide was added to dissolve the purple formazan precipitate by shaking. The Bio-Rad micro plate reader by Bio-Rad Life Medicine Products (Shanghai, China) Co., Ltd., were employed to measure the absorbance value at 490 nm. The tumor inhibitory rates of the samples were calculated according to the optical density values. Samples and standards were analysed in triplicate.

### 2.6. Tumor Inhibition Experiments In Vivo

The animal experiments were carried out by Zhejiang Academy of Medical Sciences, China. All operations were in accordance with relevant experimental animal rules and ethical guidelines. The tumor cell culture was performed as described above. The cultured cells were digested and formed a cell suspension, and then, they were harvested by centrifuging at 1200 rpm and 6 min. the collected cells were diluted to 8.0 × 10^9^ cell/L with appropriate amount of serum-free medium. Male hairless mice (HRS/J) (4-weeks old, 40 mice), weighing 20 ± 1 g, were inoculated subcutaneously in the right axilla of nude mice with 250 µL tumor cells per mouse (2 × 10^6^ cells), and then individually cultured with the blank group mice under the same conditions (22 ± 2 °C, 55 ± 5% humidity, maintained on a 12-h light/12-h dark cycle with free access to food and water). Eight mice in each group were treated in parallel. After 9 days of culture, the average tumor size of the mice reached 100 mm^3^. The drug was firstly administered by tail vein injection as the dose (0.2 mL) of PT 5 mg/kg in PT-Cas and HA-PT-Cas groups. The model group was given the same volume of normal saline as the other drug groups. All animals administered with the treatments twice a week (Monday and Thursday), and feeding was terminated on the third day after 7 times of administration.

### 2.7. Statistical Analysis

Statistical differences between two groups were evaluated using Student’s *t* test. A probability value of <0.05 was considered significant. The data are presented as mean ± standard deviation of three determinations. All the experiments were performed and analyzed in triplicate.

## 3. Results and Discussion

### 3.1. Optimized Preparation of HA Coated PT-Entrapped Casein NPs

Casein as an excellent nanocarriers has been widely used for hydrophobic drugs and bioactive compounds due to its emulsification [19]. However, our previous work indicated that most of the drug molecules leaked out easily through the irregular channels and inner cavities of casein NPs [5], especially when the samples were heated to around 37 °C, which was close to body temperature. Therefore, this work explored the use of biocompatible hyaluronic acid oligosaccharides for coating the NPs to enhance biological activity. As show in Figure 1, PT-loaded casein NPs were prepared by adding various concentrations of caseins. The effects on the entrapment efficiency of paclitaxel showed that with nanocarrier caseins increase of 10–30 mg/mL, drug molecules resulted in the obvious improvement in loading efficiency with the maximum of 71.3%, but no further increase occurred at higher casein concentration (Figure 1a). Furthermore, adequate encapsulation on the surface of paclitaxel-casein NPs had an important effect on high encapsulation efficiency. Therefore, the cross-linking reaction time of HA and PT-loaded casein NPs was investigated. Compared with the control group, the encapsulation efficiency of paclitaxel-casein NPs integrated with HA oligosaccharides was significantly improvement with increase crosslinking reaction time. Results showed that the encapsulation efficiency of paclitaxel-casein NPs was further increased from 71.3% to 93.1% with increasing crosslinking reaction time up to 120 min (Figure 1b). This finding indicates that sufficient cross-linking reaction time is favorable for hyaluronic acid to adhere firmly to the surface of NPs and form more stable nanocomplexes. 

To further improve the drug loading of HA-PT-Cas nanocomposites, the effects of the concentration of paclitaxel and various ratios of HA to casein were investigated. A significant increasing trend of the loading efficiency was observed with increasing paclitaxel amounts from 1.6 mg/mL to 4.0 mg/mL, resulting in a maximum 2.1-fold increase. By increasing the amount of HA (HA-Cas ratio from 1:1 to 4:1), the drug loading of NPs reached the maximum values of 0.095, 0.080, and 0.1297 mg/mg casein by using 2.8, 3.2, and 4.0 mg/mL paclitaxel, respectively (Figure 1c). Nevertheless, the NPs dispersion generated turbidness and precipitates (Figure 1d) when paclitaxel concentration exceeded 3.2 mg/mL. This finding indicated the influence of a high loading amount of paclitaxel on the stability of NPs. Therefore, the preparation processes of HA-PT-Cas were optimized as 2.8 mg/mL paclitaxel and the HA-Cas ratio of 3:1.

The physical and chemical parameters (i.e., the particle size, polydispersity, and zeta potential) are important characteristic indexes, because they have direct relevance to the stability of NPs [20], sustained-release efficiency of active compound, and permeability of nanocomplexes [17]. As shown in Table 1, the effects of HA/casein values and paclitaxel concentrations on the particle size, polydispersity, and zeta potential of HA oligosaccharide-encapsulated paclitaxel-casein NP (HA-PT-Cas NPs) were further studied. The particle size of HA-PT-Cas NPs was showed a descending trend with increasing HA/Cas ratio from 1:1 to 4:1, which resulted in the formation of slightly smaller NPs from 283.9 nm to 226.1 nm with the addition of 2.8 mg/mL paclitaxel. However, the addition of initial paclitaxel concentration affected the size of NPs, which resulted in the formation of larger NPs. The zeta potential values of HA oligosaccharides modified paclitaxel-casein nanoparticles were distributed in the range of −8.32 to −17.17 mV due to the presence of the ionized carboxylic group of HA in the shell. The maximum absolute potential of 17.17 mV (potential −17.17 mV) was obtained with 2.8 mg/mL paclitaxel and the HA-Cas ratio of 3:1. The zeta potential values of the particles were negatively charged and slightly higher than those of previous reported HA-ceramide nanoparticles (–14.76 to –28.62 mV) [21]. Zeta potential is also an important parameter that reflects the physicochemical and biological stabilities of NPs in suspension. Higher absolute value of zeta potential helps the formulation enhance the long-term stability and monodispersity [17]. The polydispersity index of NPs was found to be in the range of 0.101–0.170, which indicated the relatively good size uniformity of HA-PT-Cas nanoparticles. NP size is a factor determining the in vivo fate, including the retention time and bio-distribution. The narrow size distribution indicated the formation of a uniform structure owing to our narrow-spectrum of specific molecular weight HA oligosaccharides prepared by leech hyaluronidase [12].

### 3.2. Stability Assessment of HA-PT-Cas NPs

The storage stability study of HA-PT-Cas NPs was evaluated under different conditions. To determine the most effective stabilization, we prepared PT-Cas, HA-PT-Cas (HA/Cas 3:1), and HA-PT-Cas (HA/Cas 4:1) NPs with different amounts of HA and then incubate them at 37 °C for 180 min (Figure 2a). The stability tests showed that the encapsulation efficiency of PT-Cas NPs without modification decreased significantly from 71.5% to 34.1% after incubation for 30 min, and only 20.3% paclitaxel remained after 180 min. However, HA-PT-Cas coated with HA showed remarkably stability, and about 90.1% (HA/Cas 3:1) and 87.3% (HA/Cas 4:1) paclitaxel still remained after incubation for 30 min. Especially after incubation for 180 min, HA-PT-Cas (HA/Cas 3:1) still retained 64.7% paclitaxel, and the NP dispersion was still clear, whereas PT-Cas generated severe turbidness due to the large amount of paclitaxel precipitates (Figure 2b). These results indicated that the coating of HA in the outer layer plays a key role in stabilizing the PT-Cas NPs. The storage stability of HA-PT-Cas NPs when incubated at 4 °C was investigated (Figure 2c). HA-PT-Cas was extremely stable for 12 days, with 81.4% of paclitaxel remaining. However, PT-Cas NP solution without HA modification began to gradually decrease at 4 °C after 16 h with only 27.8% of remaining paclitaxel, and the instability of the PT-Cas NPs resulted in a loss of about 84% of paclitaxel after 288 h. 

The freeze-dried storage stability is also an important property of nanoparticles. Figure 2d shows that the lyophilization of HA-PT-Cas had little effect on the characterization compared with PT-Cas NPs. HA-PT-Cas NPs were stable for more than 30 days with no loss of encapsulation efficiency, whereas the stability of PT-Cas gradually decreased after 30 days with 18.7% loss of encapsulation efficiency. Furthermore, the particle sizes of NPs were further investigated, and results showed that PT-Cas after lyophilization presented a gradual decrease of particle sizes from 99.2 nm to 80.2 nm during the 30-day storage. Although these values presented a slight decrease by 8 nm, HA-PT-Cas NPs exhibited remarkable physical and chemical stability in freeze-dried storage for 30 days. The stability advantages of HA-coated PT-Cas NPs can prolong the circulation time of PT and increase the targeted accumulation of drug in tumors due to the specific binding of HA to CD44 [15]. Tumor progression frequently coupled with hyaluronidase-mediated turnover [22], which facilitates the degradation of HA-coated on the outer shell of targeted HA-PT-Cas NPs, thus facilitating the effective sustained-release of drug molecules.

### 3.3. Characterization of Nanoparticles

The morphology of the PT-loaded bovine casein NPs was observed by a transmission electron microscope (TEM). Individual PT-Cas NPs exhibited spherical casein micelles with an average size of 100 nm (Figure 3a). The morphology of HA-PT-Cas with a molecular weight of 10,000 Da was shown in Figure 3b. The slight swelling of HA-PT-Cas NPs results in the larger size of particles with an average size of 230 nm, which indicated that HA was successfully coated on the surface of PT-Cas by the cross-linking reaction. Additionally, the molecular weight of HA had a great influence on particle size due to the macromolecule viscosity of HA. Therefore, the 500 kDa HA was also used to modify PT-Cas NPs, resulting in larger particle size and turbidness. These results also reflected that the larger size of particles observed by TEM might be a result of swelling and aggregation of the HA hygroscopic properties during dispersion in an aqueous medium [23]. This result is consistent with those of a previous study on lysozyme-coated eugenol-casein nanoparticles [24], and other self-assembled nanoparticles based on HA derivatives were >200 nm [25]. Micelles with a mean diameter <200 nm are susceptible to renal excretion during long circulation, whereas those with a mean diameter of <5 mm have access to small capillaries [26]. Therefore, many approaches are being employed to enhance the resistance of renal clearance of the carrier for its properties to better suit the incorporated drug. In this work, HA oligosaccharides modified the outer layer of PT-Cas NPs (~230 nm), which was due to the steric hindrance caused by the presence of a hydrophilic shell of polymer micelles. The rapid loss of paclitaxel drug molecules from micellar cavities was prevented.

FT-IR analysis spectra were carried out to investigate the chemical structure and interaction between casein NPs and HA oligosaccharides. As shown in Figure 3c, the characteristic absorption bands of casein were 3371.76 cm^−1^ (stretching vibration of N-H bond), 1657.05 cm^−1^ (stretching vibration of –C=O bond which belongs to –CO–NH– on casein backbone), 1530.35 cm^−1^ (due to the stretching vibration of the C–N bending in carboxylate groups), and 1077.65 cm^−1^ (C–O–H bending) [27,28,29]. The characteristic bands observed for HA oligosaccharides were 3420 cm^−1^ (the intra-and intermolecular stretching vibration of –OH and N–H groups), 2927 cm^−1^ (the symmetric stretching vibration of –CH_2_ group), 1635 cm^−1^ and 1408 cm^−1^ (the symmetric and asymmetric vibration of COO− group), and 1045 cm^−1^ (correlated with the C–O–C hemiacetalic system saccharide units) [30,31]. FT-IR spectra of HA-PT-Cas NPs showed the similar characteristic peaks as well as HA oligosaccharides but with the obvious decreased absorbance of the major absorption peaks. It also clearly indicates that the presence of both HA and the casein protein in the nanoparticles; the determinative bands of amide I (1645 cm^−1^) and II (1531 cm^−1^) as well as the vibrations of COO– and C-O(H) are presented [32]. This may be due to the HA-coated PT-Cas nanostructure resulted in the overlapping of stretching vibration from the derivative structure with these groups from the HA structure. Similar results have been reported in HA/maleic anhydride derivatives and octenyl succinic anhydride–modified HA derivatives [31,33].

### 3.4. Antitumor Assessment of HA-PT-Cas NPs

Excellent stability of NPs can provide satisfactory sustained-release effect for drug molecules, to avoid the rapid release of drug molecules in a short period of time, which makes it difficult to maintain long-term efficacy. Hereby we studied the anti-tumor effects of HA-PT-casein micelle dispersions on the cell viability of human malignant melanoma cells (A375) and compared these effects with those of PT-casein micelle dispersions. The anti-tumor effects of HA-PT-Cas, PT-Cas, and cis-platinum against A375 cells were shown in Figure 4. The uncoated PT-Cas NPs showed the lowest efficiency of 47.81% in inhibiting A375 cells, whereas the HA encapsulated and stable HA-PT-Cas exhibited the distinctly higher efficiency in inhibiting A375 cells for 20 μg/mL concentrations (Figure 4a), which was comparable with the 64.93% antitumor efficiency of 10 μg/mL cis-platinum. Furthermore, the morphology of A375 tumor cells treated with the cis-platinum and PT NPs were showed in Figure 4b. Compared with cisplatin, the cells treated with PT NPs showed aggregative apoptosis, and the number of cells treated with HA-PT-Cas NPs was significantly lower than that of the control and PT-Cas groups. The HA-Cas NPs showed slight antitumor activity of 7.29% due to the extremely low dose and in vitro treatment. The important outcome of hyaluronan turnover is the formation of low molecular weight products that induce angiogenesis [34]. However, details of the underlying mechanisms by which hyaluronan turnover influences tumor cell behavior are still unclear [13].

To further verify the antitumor efficacy of HA-coated PT-casein-NPs in vivo, mice implanted with A375 tumor cells were intravenously administered with the dispersions of HA-PT-Cas and PT-Cas NPs via the caudal vein every 3 days for 24 consecutive days. Compared with the control group, the tumor volume results indicated that paclitaxel NPs showed a strong tumor inhibition effect (Figure 5a). The tumors treated with HA-coated PT-Cas NPs were significantly smaller than those treated with PT-Cas. Meanwhile, the final tumor weights after sacrifice further demonstrated that the tumor inhibition rate reached 74.6% for mice supplemented with HA-PT-Cas NPs (5 mg PT/kg b.wt.). In contrast, the tumor inhibition rate of PT-casein NPs without HA-coated was 39.8% for mice treated with 5 mg PT/kg b.wt., which was significantly lower than that of HA-PT-Cas NPs (Figure 5b). HA coated on the casein nanoparticles showed higher antitumor efficacy and fewer side effects than PT-Cas NPs in mice. The mechanisms may be the following. (1) The excellent stability of HA-PT-Cas NPs delayed PT release, thereby prolonging the drug circulation time and finally concentrating the drug in tumor tissues. (2) HA coated on the outer layer was biocompatible and biodegradable; it was specifically degraded by hyaluronidase coupled with tumor progression, thereby leading to the generation of lower molecular weight (LWM) oligosaccharides and the release of NP drugs in the tumor tissue. (3) LMW HA oligosaccharides were specifically recognized by CD44 on the surface of tumors, resulting in more targeted intracellular uptake of PT in the tumor cell. These data clearly indicated that HA-PT-Cas NPs have potential for the treatment of tumor growth. 

## 4. Conclusions

Low-molecular-weight HA oligomers were coated on the surface of casein NPs via chemical conjugation to increase the stability and bioactivity of paclitaxel-loaded PT-Cas NPs. Compared to uncoated PT-Cas NPs, the novel formulated HA-PT-Cas NPs (HA/casein ratio of 3:1) could reach a higher entrapment efficiency (93.1%) and exhibited satisfactory stability. The antitumor effects assay confirmed that HA-PT-Cas NPs showed higher cytotoxicity in vitro and better antitumor efficacy in vivo. Furthermore, HA oligomers showed slight antitumor activity with low dose. Therefore, HA coated on the casein NPs showed higher bioavailability for drug delivery, which was mainly due to the fact that the block of NP cavity prolonged the drug circulation time, the specific recognition of tumors increased drug targeted accumulation, and the specific degradation by hyaluronidase coupled with tumor progression ensured targeted release. This strategy of the structural modification of casein NPs by using HA oligomers is expected to be effective for loading other hydrophobic drugs. 

## Figures and Tables

**Figure 1 nutrients-14-03888-f001:**
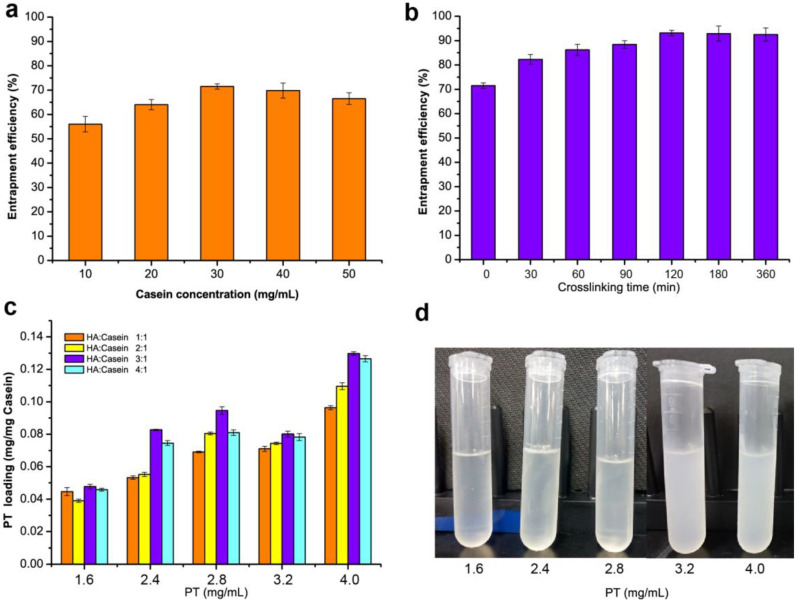
**Evaluation of nanoparticles formulation with various concentrations of casein and hyaluronic acid oligosaccharides.** (**a**) Effect of casein concentrations on entrapment efficiency. (**b**) Analysis of the entrapment efficiency of nanoparticles with crosslinking time of HA and casein. (**c**) Effects of paclitaxel (PT) concentrations and the HA/casein values on the PT loading of nanoparticles. (**d**) Effect of the PT concentrations on the turbidness of nanoparticles.

**Figure 2 nutrients-14-03888-f002:**
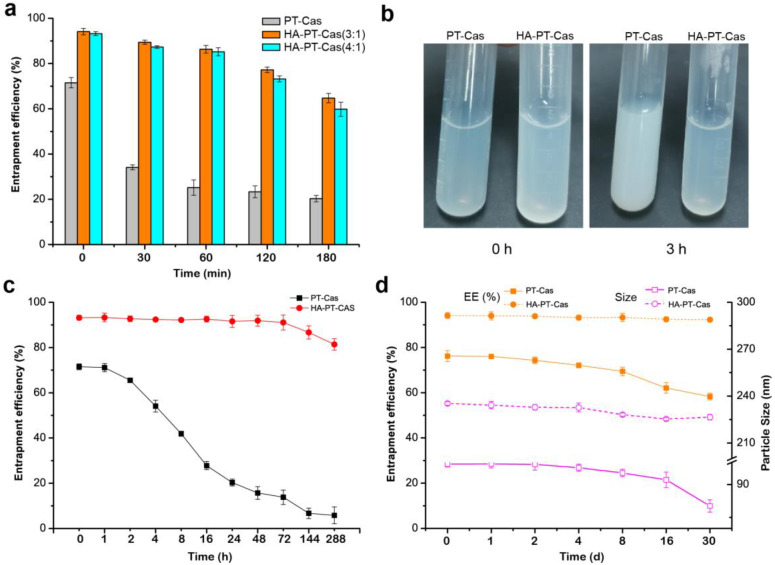
**Investigation of the stability of HA-PT-Cas nanoparticles.** Effects of the HA/casein values on the stability of entrapment efficiency (**a**) and turbidness (**b**) of nanoparticles incubated in 37 °C. (**c**) The entrapment stability of PT-Cas and HA-PT-Cas nanoparticles stored at 4 °C for 288 h; (**d**) Effects of lyophilization storage for 30 days and redissolution on the entrapment efficiency and particle size of nanoparticles.

**Figure 3 nutrients-14-03888-f003:**
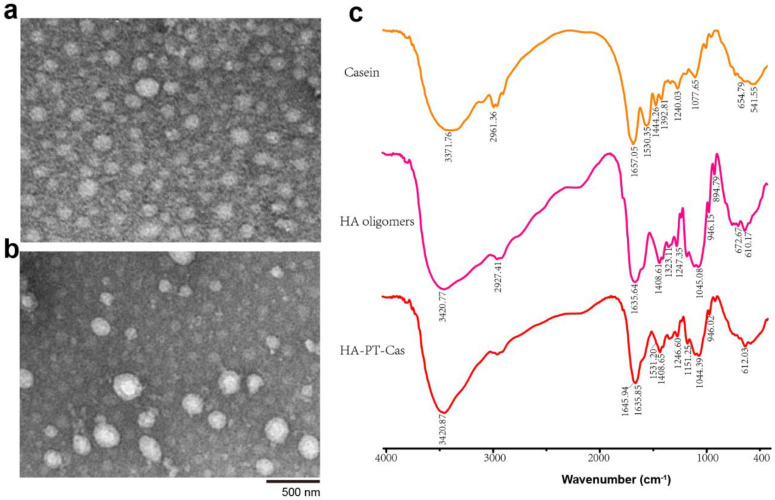
Transmission electron microscopic images of nanoparticles of PT-Cas (**a**) and HA-PT-Cas (**b**); FT-IR spectra of casein, HA oligomers and HA-PT-Cas (**c**).

**Figure 4 nutrients-14-03888-f004:**
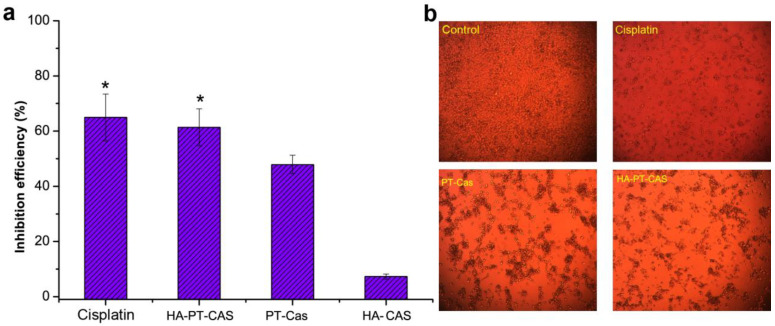
**Assessment of antitumor activity of PT nanoparticles on tumor cells.** (**a**) The inhibition effects of cis-platinum and PT nanoparticles on A375 tumor cells. (**b**) The morphology of tumor cells treatment with the cis-platinum and PT nanoparticles. ** p* < 0.05, significantly different compared with the PT-Cas group.

**Figure 5 nutrients-14-03888-f005:**
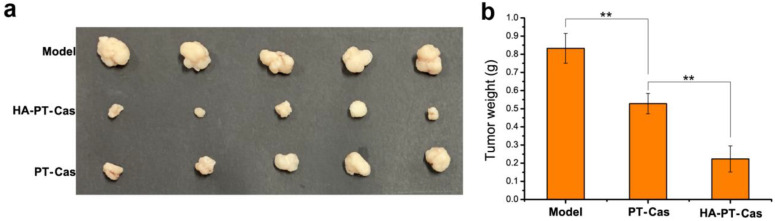
**The antitumor activity of animal experiments with different PT nanoparticles treatments**. (**a**) Changes in tumor volumes in model groups, and HA-PT-Cas NPs and PT-Cas NPs groups after the final drug treatments. (**b**) Tumor weight changes. ** indicate statistical difference of the values at *p* < 0.01.

**Table 1 nutrients-14-03888-t001:** Effects of PT concentrations and the HA/casein values on the particle size, PDI and zeta potential of NPs.

PT Concentration (mg/mL)	HA:Cas	Size (nm)	PDI	Zeta (mV)
1.6	1:1	243.5 ± 4.8	0.140 ± 0.013	−9.99 ± 0.47
	2:1	276.6 ± 12.1	0.107 ± 0.005	−8.76 ± 0.23
	3:1	217.2 ± 5.2	0.122 ± 0.007	−13.2 ± 0.51
	4:1	236.2 ± 7.1	0.103 ± 0.011	−8.32 ± 0.27
2.4	1:1	253.4 ± 4.1	0.142 ± 0.008	−10.09 ± 0.68
	2:1	261.3 ± 5.6	0.128 ± 0.006	−10.13 ± 0.92
	3:1	235.3 ± 2.6	0.112 ± 0.012	−15.05 ± 0.53
	4:1	229.0 ± 3.9	0.107 ± 0.009	−7.33 ± 0.78
2.8	1:1	283.9 ± 5.3	0.117 ± 0.010	−12.09 ± 0.62
	2:1	248.0 ± 7.2	0.143 ± 0.013	−13.13 ± 0.92
	3:1	235.8 ± 3.1	0.101 ± 0.006	−17.17 ± 0.83
	4:1	226.1 ± 2.5	0.131 ± 0.009	−11.08 ± 0.91
3.2	1:1	276.7 ± 9.7	0.170 ± 0.014	−9.81 ± 0.35
	2:1	249.5 ± 3.4	0.122 ± 0.009	−9.88 ± 0.71
	3:1	268.0 ± 4.8	0.143 ± 0.007	−10.08 ± 0.69
	4:1	261.5 ± 11.3	0.127 ± 0.015	−9.54 ± 0.93
4.0	1:1	296.2 ± 6.7	0.121 ± 0.011	−9.27 ± 0.79
	2:1	279.3 ± 4.2	0.133 ± 0.008	−9.69 ± 0.57
	3:1	263.0 ± 3.1	0.135 ± 0.015	−10.11 ± 0.66
	4:1	251.8 ± 7.9	0.149 ± 0.012	−9.67 ± 0.83

## Data Availability

Not applicable.

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
