# Peer review of "Hyaluronan Oligosaccharides-Coated Paclitaxel-Casein Nanoparticles with Enhanced Stability and Antitumor Activity"

_nutrients, 2022, doi:10.3390/nu14193888_

Round 1
Reviewer 1 Report
The article Hyaluronan oligosaccharides-coated paclitaxel-casein nanoparticles with enhanced stability and antitumor activity is an interesting example of the use of both casein and hyaluronic acid as a carrier for anti-cancer drugs. In the descriptions of individual studies, the number of repetitions was not given, which requires supplementing. Besides, there are a few editorial mistakes, e.g. line 40-41 pol-ymer. In my opinion, the work can be printed after minor corrections
Author Response
The article Hyaluronan oligosaccharides-coated paclitaxel-casein nanoparticles with enhanced stability and antitumor activity is an interesting example of the use of both casein and hyaluronic acid as a carrier for anti-cancer drugs.
Response 1: We sincerely thank reviewer #1 for the positive comments and suggestions on our work. As suggestion, we have tried our best to carefully revise to improve the quality of the manuscript, which we hope meet with approval.
In the descriptions of individual studies, the number of repetitions was not given, which requires supplementing.
Response 1: Thank the reviewer #1 for this good suggestion; as suggestion, we have added this description in revised manuscript.
Besides, there are a few editorial mistakes, e.g. line 40-41 pol-ymer. In my opinion, the work can be printed after minor corrections
Response 1: Thank the reviewer #1 for this careful observation; we have revised it and carefully checked out these minor errors through the manuscript.
Reviewer 2 Report
The manuscript reports the study on the development hyaluronic acid oligosaccharides-coated paclitaxel-loaded casein nanoparticles. This study is aimed at solving the important problem of encapsulation of hydrophobic substances. However, in my opinion, there are some aspects that should be corrected:
- Page 2, introduction – the authors should add information about paclitaxel. What problem are the developed systems solving like paclitaxel delivery systems?
- Page 3, line 117 - please indicate how the sample preparation for FTIR was carried out
- Page 3, line 125 – please provide information about detector in HPLC system.
Author Response
The manuscript reports the study on the development hyaluronic acid oligosaccharides-coated paclitaxel-loaded casein nanoparticles. This study is aimed at solving the important problem of encapsulation of hydrophobic substances. However, in my opinion, there are some aspects that should be corrected:
Response 2: We sincerely thank reviewer #2 for the positive comments and suggestions on our work. As suggestion, we have tried our best to carefully revise to improve the quality of the manuscript, which we hope meet with approval.
Page 2, introduction – the authors should add information about paclitaxel. What problem are the developed systems solving like paclitaxel delivery systems?
Response 2: Thank the reviewer #2 for this good suggestion; as suggestion, we have added these descriptions about paclitaxel in Introduction section.
Page 3, line 117 - please indicate how the sample preparation for FTIR was carried out
Response 2: Thank the reviewer #2 for this good suggestion; as suggestion, we have added the description of sample preparation for FTIR in revised manuscript.
Page 3, line 125 – please provide information about detector in HPLC system.
Response 2: Thank the reviewer #2 for this good suggestion; as suggestion, we have added the information of detector in revised manuscript.
Reviewer 3 Report
The manuscript "Hyaluronan oligosaccharides-coated paclitaxel-casein nanoparticles with enhanced stability and antitumor activity" reports the formulation, characterization and antitumoral activity of HA coated PT-entrapped casein NPs. The study is interesting, well organized and the results are clearly presented.
Only some changes are suggested. It could be useful to add a a figure with a diagram to illustrate the preparation of the NP. The text of each section, which is well written, should be divided in more paragraphs. For example at line 244 the phrase that starts with "The freeze-dried" should start at the next line. This suddivision of section in paragraphs should make. reading more fluent
Author Response
The manuscript "Hyaluronan oligosaccharides-coated paclitaxel-casein nanoparticles with enhanced stability and antitumor activity" reports the formulation, characterization and antitumoral activity of HA coated PT-entrapped casein NPs. The study is interesting, well organized and the results are clearly presented.
Response 3: Firstly, we sincerely thank reviewer #3 for the positive comments and suggestions on our work. As suggestion, we have tried our best to carefully revise and polished the manuscript by two native English speakers with technical knowledge in the area, to improve the quality of the manuscript which we hope meet with approval.
Only some changes are suggested. It could be useful to add a figure with a diagram to illustrate the preparation of the NP.
Response 3: Thank the reviewer #3 for this good suggestion. We have showed a diagram to illustrate the preparation of the NP in graphical abstract (the following figure), and we have supplemented this “graphical abstract” file in revised manuscript.
The diagram of the HA-PT-Cas NP preparation
The text of each section, which is well written, should be divided in more paragraphs. For example at line 244 the phrase that starts with "The freeze-dried" should start at the next line. This subdivision of section in paragraphs should make. reading more fluent
Response 3: Thank the reviewer #3 for this good suggestion. As suggestion, we have revised this subdivision of section to make it more fluent.